The clinical value of carcinoembryonic antigen for tumor metastasis assessment in lung cancer

Wang Jiasi 1
Chu Yanpeng 2
Li Jie 2
Wang Tingjie 1
Sun Liangli 1
Wang Pingfei 2
Fang Xiangdong 3
Zeng Fanwei 2
Wang Junfeng j.wang@lumc.nl 4
Zeng Fanxin zengfx@pku.edu.cn 2 3
1 Department of Clinical Laboratory, Dazhou Central Hospital , Dazhou , Sichuan , China
2 Department of Clinical Research Center, Dazhou Central Hospital , Dazhou , Sichuan , China
3 Department of Oncology, Dazhou Central Hospital , Dazhou , Sichuan , China
4 Department of Biomedical Data Sciences, Leiden University Medical Center , Leiden , The Netherlands
Maddi Abhiram
Electronic publication date: 2019 Aug 7
Publication date: 2019
Volume: 7
Electronic Location ID: e7433
Received 2019 Apr 18; Accepted 2019 Jul 8
Copyright: ©2019 Wang et al.
Copyright year: 2019
Copyright holder: Wang et al.
License: This is an open access article distributed under the terms of the Creative Commons Attribution License, which permits unrestricted use, distribution, reproduction and adaptation in any medium and for any purpose provided that it is properly attributed. For attribution, the original author(s), title, publication source (PeerJ) and either DOI or URL of the article must be cited.
License URL: https://creativecommons.org/licenses/by/4.0/

Keywords: Serum carcinoembryonic antigen, Lung cancer, Tumor metastasis, Performance of assessment, Receiver operating characteristic

Funding: Scientific Research Fund of Sichuan health and Health Committee 18PJ361 Scientific Research Fund of Technology Bureau in Sichuan Province 2018138 2018JY0324 This work was supported by the Scientific Research Fund of Sichuan health and Health Committee (No. 18PJ361) and the Scientific Research Fund of Technology Bureau in Sichuan Province (No. 2018138, No. 2018JY0324). The funders had no role in study design, data collection and analysis, decision to publish, or preparation of the manuscript.

==============================
Background

Carcinoembryonic antigen (CEA) as a diagnostic or prognostic marker has been widely studied in patients with lung cancer. However, the relationship between serum CEA and tumor metastasis in lung cancer remains controversial. This study aimed to investigate the ability of serum CEA to assess tumor metastasis in lung cancer patients.

Methods

We performed a retrospective analysis of 238 patients diagnosed with lung cancer from January to December 2016 at pneumology department of Dazhou Central Hospital (Dazhou, China). Serum CEA levels were quantified in each patient at the time of diagnosis of lung cancer. Metastasis was confirmed by computed tomography (CT), and/or positron emission tomography (PET) and/or surgery or other necessary detecting methods.

Results

Of the 213 patients eligible for final analysis, 128 were diagnosed with metastasis and 85 were diagnosed without metastasis. Compared to non-metastatic patients, the serum CEA was markedly higher in patients with metastasis (p < 0.001), and the area under the curve (AUC) was 0.724 (95% CI [0.654–0.793]). Subsequent analyses regarding the number and location of tumor metastases showed that CEA also had clinical value for multiple metastases versus single metastasis (AUC = 0.780, 95% CI [0.699–0.862]) and distant metastasis versus non-distant metastasis (AUC = 0.815, 95% CI [0.733–0.897]). In addition, we found that tumor size, histology diagnosis, age and gender had no impact on the assessment performance of CEA.

Conclusion

Our study suggested the serum CEA as a valuable marker for tumor metastases assessment in newly diagnosed lung cancer patients, which could have some implications in clinical application.

Introduction

Lung cancer is the most common cause of death among all cancers worldwide (Torre et al., 2015; Wang, Song & Zhang, 2018). The overall survival rate depends on the stage of lung cancer, and patients with advanced lung cancer commonly have poor prognosis (Langer et al., 2010). Unfortunately, up to 70% of patients are already in advanced stage at the time of diagnosis (Arrieta et al., 2009; Patz Jr et al., 2007). Evidence suggests that tumor metastasis reflects the relatively advanced stage of lung cancer and is responsible for more than 70% of patient deaths (Langer et al., 2010). The two main types of lung cancer are small-cell lung cancer (SCLC) and non-small-cell lung cancer (NSCLC), and the latter consists of three subtypes: adenocarcinoma, squamous cell carcinoma and large-cell carcinoma (Zheng, 2016). In NSCLC patients, the main locations for tumor metastasis are lymph gland, bone, brain and liver (Wood et al., 2014). In NSCLC patients with no more than five metastases, systemic therapy, high-dose fractional radiotherapy or surgical removal of all affected sites can result in no progress in 3 years in 13% of patients, even in stage IV patients, who can benefit from radical therapy. At the same time, minimally invasive surgery, palliative surgery, radiotherapy and bone modifier therapy can create better prognosis in NSCLC metastasis (Reck et al., 2014). Therefore, early detection of the presence and location of metastasis is helpful for choosing treatment methods and to improve the prognosis of patients.

Clinically, metastasis is confirmed by using a combination of clinical symptoms and imaging evidence when complete pathological evidence is available for lung cancer diagnosis (Gaga et al., 2013). The most commonly employed imaging modalities include computed tomography (CT), fused-positron-emitting-tomography-CT (PET-CT), magnetic-resonance-imaging (MRI) and chest X-ray (CXR) (Reck et al., 2014). However, the high expense of inspection and other factors may cause a great burden on the patient that could hinder the clinical monitoring and early detection of lung cancer metastasis (Patz Jr et al., 2007; Zhang et al., 2017). Which, to a certain extent, limits the choice of treatment and negatively affects the prognosis of progressive cancer patients. In addition, patients can have metastases in certain areas, without obvious clinical symptoms, which are often ignored by both patients and doctors. Therefore, diagnostic techniques that are both economic and uncomplicated are urgently needed in clinics to indicate whether tumor metastases have occurred, which could guide doctors on whether a lung cancer patient with symptoms of suspected metastasis need more detailed examination.

Blood-based biomarkers can be accessed easily, quickly, and economically, so they have the potential to greatly improve the efficiency of assessment. Carcinoembryonic antigen (CEA) is a glycoprotein involved in cell adhesion, which was first found in 1965 in the blood of patients with colon cancer (Crone-Munzebrock & Carl, 1990). Previous studies have reported that CEA could influence either epithelial cells or the surrounding stromal cells and immunity to alter related signaling programs such as TGF-βR1, apoptosis regulating proteins, PI3K and AKT activities to support metastasis progression in colorectal cancer patients (Beauchemin & Arabzadeh, 2013). The level of serum CEA expression and the trend of its changes in the treatment process have been validated by several studies on the clinical value of assessing the efficacy and prognosis of NSCLC (Ardizzoni et al., 2006; Arrieta et al., 2013). High levels of tumor markers at baseline, such as CEA and CA125, are correlated with worse survival in stage III-IV NSCLC patients (Cedres et al., 2011). Additionally, some studies have revealed that increased CEA levels were associated with increased risk of developing brain metastasis in patients with advanced NSCLC (Wang et al., 2017). However, many studies have reported that the prognostic value of CEA for lung cancer patients was ambiguous (Tsoukalas et al., 2017). Therefore, further studies are required for the diagnostic value of testing serum CEA in lung cancer patients.

We retrospectively analyzed the relationship between CEA and tumor metastasis in newly diagnosed patients with NSCLC and SCLC, to investigate the relationship between CEA and tumor metastasis, and assess whether CEA has certain clinical guiding value for organs involved and the number of organs involved in tumor metastasis.

Materials and Methods

Study cohort

We retrospectively reviewed 238 patients with a diagnosis of lung tumors from January to December 2016 at the pneumology department of Dazhou Central Hospital in this study. Clinical information, including gender, age, metastatic locations, histological type and tumor size was retrospectively obtained from electronic medical records. All these patients underwent fiberoptic bronchoscopy biopsy at the initial stage of cancer diagnosis after hospitalization. The specimens needed for pathological diagnosis came from fiberoptic bronchoscopy, percutaneous lung biopsy, or surgical resection. The histological type was confirmed by pathological morphology and immunohistochemistry. The study was approved by the medical ethics review board of Dazhou Central Hospital (Ethical Application Ref: IRB00000002-17002). The medical ethics review board waived the need for informed consent from participants in this study.

The exclusion criteria were as follows: (i) patients who lacked serum CEA data (n = 7); (ii) patients diagnosed with lung cancer, but without a pathological histology basis (n = 18). Finally, 213 patients were included in the analyses.

Index test

Serum CEA level was detected at the initial stage of cancer diagnosis, using a CEA test kit (Roche Diagnostics Corp, Shanghai, China) by cobas e601 analyzer (Werner et al., 2016). The experimental operation was carried out according to the reagent protocol. The standard cut-off value of CEA was 5.0 ng/mL, as recommended by the manufacturers of the assay kits.

Definition of groups

Metastasis group: The presence of metastasis was confirmed within the first period of admission diagnosis (no more than one month). Patients diagnosed with lung cancer and with obvious images of metastasis in imaging were included in the metastasis group.

Non-metastasis group: Patients with lung cancer but without detected any lymph node metastasis, intrapulmonary metastasis, and any other metastasis within the first hospitalization period and metastases detected beyond the first hospitalization period were included in the non-metastasis group.

Single metastasis group: Single metastasis is for a single organ, not for single site. Patients with a single metastasis site and multiple metastases sites in one single organ were all included in the single metastasis group.

Multiple metastases group: Patients with multiple organ metastases were included in the multiple metastases group.

Non-distant metastasis group: Patients with simple mediastinal/hilar lymph nodes and/or intrapulmonary metastasis were included in the non-distant metastasis group.

Distant metastasis group: Patients excluded from the non-distant metastasis group were included in the distant metastasis group.

Reference standard

In our study, the diagnosis of metastasis was based on the complete pathological evidence in the diagnosis of lung cancer, combined with the patient’s clinical characteristics and imaging evidence (if necessary, this was combined with pathological examination and/or the specific expression level of tumor markers, such as the confirmation of partial lymph node metastasis by lymph node dissection, the existence of pathological evidence of pleural effusion and pericardial effusion, and the high expression of CA125 in pleural effusion, and other diagnostic criteria for metastasis confirmation). All metastases, not detected during the first hospitalization period (no more than one month), were not considered to have metastases. CT, PET, surgery, MRI and Fluorodeoxyglucose (FDG)–PET-CT scan are common imaging modalities for metastasis assessment (Reck et al., 2014).

Statistical analysis

Continuous variables were expressed as median (IQR), while categorical variables were expressed with count and percentage. The Kruskal-Wallis test was used to compare the age difference between patients with and without metastases. The Pearson Chi-squared test was used to compare gender, histology and size of tumor difference between patients with and without metastases. The Mann–Whitney U test was used to compare CEA levels between patient groups. The relation between CEA level and metastasis was visualized using restricted cubic spline. Receiver operating characteristic (ROC) curve and area under the curve (AUC) were used to represent the overall test accuracy. The optimal cut-off value was determined using Youden’s Index. Sensitivity and specificity of the index test were reported along with the Clopper–Pearson confidence intervals. Covariate adjusted ROC curve was generated using a percentile values approach (Janes & Pepe, 2009). MedCalc and R (version: R 3.4.3 for windows (x64), R Core Team, 2017) were used for statistical analysis, and a p-value < 0.05 was considered statistically significant.

Results

Demographics

There were 213 patients eligible for the final analyses based on the exclusion criteria, 170 male and 43 female, with a median age of 62 years (range from 35 to 82 years). A cohort diagram was shown in Fig. 1 and the patients’ demographic characteristics were shown in Table 1. The distribution of lung cancer metastasis and partial imaging evidence for metastases were shown in Figs. S1 and S2.

Figure 1 Cohort diagram of the criteria for patients inclusion and exclusion.

Abbreviations: CEA, Carcinoembryonic antigen.

Table 1 Patients’ characteristics.

Characters	Lung cancer(overall)	Lung cancer(with metastasis)	Lung cancer(without metastasis)	p-value	
N	213	128	85		
Age (median(IQR))	62.0 (53.0, 68.0)	61.0 (51.0, 68.0)	63.0 (56.0, 68.0)	0.070	
Gender				0.048	
Female	43 (20.2%)	32 (25%)	11 (12.9%)		
Male	170 (79.8%)	96 (75%)	74 (87.1%)		
Histology				0.713	
NSCLC	165 (77.5%)	97 (75.8%)	68 (80%)		
Adenocarcinoma	64 (30%)	49 (38.3%)	15 (17.6%)		
Squamous cell	100 (46.9%)	47 (36.7%)	53 (62.4%)		
Large Cell	1 (0.5%)	1 (0.8%)	0 (0%)		
SCLC	38 (17.8%)	24 (18.8%)	14 (16.5%)		
Unclassified	10 (4.7%)	7 (5.5%)	3 (3.5%)		
Size of tumor (cm)				0.401	
≤3	34 (16%)	23 (18%)	11 (12.9%)		
3–5	60 (28.2%)	33 (25.8%)	27 (31.8%)		
5–7	41 (19.2%)	21 (16.4%)	20 (23.5%)		
>7	20 (9.4%)	12 (9.4%)	8 (9.4%)		
Unmeasured	58 (27.2%)	39 (30.5%)	19 (22.4%)		
Notes.

Abbreviations NSCLC Non-small cell lung cancer

SCLC Small Cell Lung Cancer

Unclassified: The pathologic diagnosis was lung cancer, but without typing.

Unmeasured: Because of special circumstances, such as emphysema, voluminous pleural effusion, tumor of bronchus, the size of the tumor could not be accurately measured.

CEA level in assessment of tumor metastasis

Among the 213 lung cancer patients with a definite diagnosis of cancer, 128 patients were included in the metastasis group and 85 patients were included in the non-metastasis group (Table 1). Serum CEA (log-transformed) was significantly higher in the metastasis group (p < 0.001, Figs. 2A–2B), and the AUC was 0.724 (95% CI [0.654–0.793], Fig. 2C). We further investigated the relation between CEA level and metastasis without predefined model structure. We found a clear trend of increase in metastasis above 5.0 ng/mL (the red line in Fig. S3). The optimal cut-off value is 4.69 ng/mL, which was comparable to the recommended cut-off value 5.0 ng/mL, so we used the recommended cut-off value to calculate the sensitivity and specificity. The sensitivity and specificity of serum CEA were 0.851 (95% CI [0.763–0.916]) and 0.597 (95% CI [0.503–0.686]) respectively. Positive predictive value (PPV) and negative predictive value (NPV) were 62.50% (95% CI [56.87%–67.81%]) and 83.53% (95% CI [75.37%–89.37%], Table 2).

Figure 2 CEA level in diagnosis of tumor metastasis.

(A) Boxplot of log(CEA) in metastatic group and non-metastatic group. (B) Distribution of log(CEA) grouped by metastasis (yes versus no). (C) Receiving operator characteristic curve(ROC) analysis based on the sensitivity and specificity of CEA on tumor metastasis.

Table 2 Diagnostic performance of CEA.

Outcome	Metastasis(all)	Number of tumors(in metastasis patients)	Location of metastasis(in metastasis patients)	
N	213	128	128	
AUC (95% CI)	0.724 (0.654–0.793)	0.780 (0.699–0.862)	0.815 (0.733-0.897)	
Cut-off (in ng/mL)	5.0	7.17	6.03	
Sensitivity (95% CI)	85.11% (76.28%, 91.61%)	76.47% (64.62%, 85.91%)	69.16% (59.50%, 77.73%)	
Specificity (95% CI)	59.66% (50.28%, 68.55%)	73.33% (60.34%, 83.93%)	95.24% (76.18%, 99.88%)	
PPV (95% CI)	62.50% (56.87%, 67.81%)	76.47% (67.67%, 83.46%)	98.67% (91.58%, 99.80%)	
NPV (95% CI)	83.53% (75.37%, 89.37%)	73.33% (63.57%, 81.25%)	37.74% (31.00%, 44.98%)	
Notes.

Abbreviations AUC Area Under Curve

PPV Positive predictive value

NPV Negative predictive value

CI confidence interval

CEA level in assessment of tumor metastasis quantity

Among the 128 patients in the metastasis group, 60 patients were included in the single metastasis group and 68 patients were included in the multiple metastases group. Serum CEA was significantly increased in the multiple metastases group when compared to patients in the single metastasis group (p < 0.001, Figs. 3A–3B), and the AUC was 0.780 (95% CI [0.699–0.862], Fig. 3C). The optimal cut-off value is 7.17 ng/mL, and when this value is used, the sensitivity and specificity of serum CEA were 0.765 (95% CI [0.646–0.859]) and 0.733 (95% CI [0.603–0.839]), respectively. PPV and NPV were 76.47% (95% CI [67.67%–83.46%]) and 73.33% (95% CI [63.57%–81.25%], Table 2).

Figure 3 CEA level in diagnosis of tumor metastasis quantity.

(A) Boxplot of log(CEA) in single metastasis group and multiple metastases group. (B) Distribution of log(CEA) grouped by number of tumor metastasis. (C) Receiving operator characteristic curve based on the sensitivity and specificity of CEA on tumor metastasis (1 versus ≥ 2).

CEA level in assessment of location of tumor metastasis

Among the 128 patients in the metastasis group, 21 patients were included in the non-distant metastasis group and 107 patients were included in as the distant metastasis group. Serum CEA was significantly higher in the distant metastasis group when compared to patients in the non-distant metastasis group (p < 0.001, Figs. 4A–4B), and the AUC was 0.815 (95% CI [0.733–0.897], Fig. 4C). The optimal cut-off value is 6.03 ng/mL, and when this value is used, the sensitivity and specificity of serum CEA were 0.692 (95% CI [0.595–0.777]) and 0.952 (95% CI [0.762–0.999]), respectively. PPV and NPV were 98.67% (95% CI [91.58%–99.80%]) and 37.74% (95% CI [31.00%–44.98%], Table 2).

Figure 4 CEA level in diagnosis of location of tumor metastasis.

(A) Boxplot of log(CEA) in distant metastasis group and non-distant metastasis group. (B) Distribution of log(CEA) grouped by location of metastasis. (C) Receiving operator characteristic curve based on the sensitivity and specificity of CEA on location of metastasis (distant versus non-distant).

Impacts of histology, tumor size, patient age and patient gender on the performance of CEA

In order to evaluate the impacts of histology, tumor size, patient age and patient gender on the performance of CEA, Crude ROC curve and ROC curve adjusted of these factors were performed.

Of the 213 patients diagnosed with lung cancer, 10 cases could not be confirmed with histology due to insufficient tissue material for histological diagnoses, while the other 203 patients had definite histological diagnosis (Table 1). In all patients with identified histology, there were 38 SCLC (38/203, 18.7%) and 165 NSCLC (165/203, 81.3%). After adjusting for histological diagnosis, the AUC did not change, thus histological diagnosis has no impact on the assessment performance of CEA (Fig. S4A).Tumor size was determined according to the greatest dimension of tumor. Because of special circumstances, such as emphysema, voluminous pleural effusion, tumor of bronchus, the size of the tumor in 58 patients (out of 213 patients) could not be accurately measured (showed in Table 1), thus leaving 155 patients eligible for this analysis. After adjusting for tumor size, the AUC did not change, thus tumor size has no impact on the assessment performance of CEA (Fig. S4B). Scatter plot of tumor size and CEA levels also showed that with the increase of the size of tumors, the increased risk of CEA did not increase (Multiple R2 = 0.001, Adjusted R2 =  − 0.005, p-value = 0.68, Fig. S5). All patients had their age and gender recorded upon receiving the CEA test, and 213 patients were included for this analysis. After adjusting for the age and gender of the patient, the AUC did not change, thus the age and gender of the patient has no impact on the assessment performance of CEA (Figs. S4C–S4D).

Discussion

In this retrospective study, we identified serum CEA as a diagnostic marker for assessing tumor metastasis in newly diagnosed lung cancer patients. Patients in the metastasis group had a significant higher CEA level versus patients in the non-metastasis group. The ROC curve demonstrated that CEA had good ability in assessing tumor metastasis. The optimal cut-off value for CEA in this dataset was 4.69 ng/mL determined by Youden’s Index, which was close to the standard cut-off value (5.0 ng/mL) supported by CEA test kit. Moreover, in patients with tumor metastasis, serum CEA was significantly increased in patients with multiple metastases compared with patients with single metastasis. A cut-off of 7.17 ng/mL could optimally distinguish between single metastasis and multiple metastases. Additionally, our data showed that CEA was also an effective marker for other metastases assessment except for mediastinal/hilar lymph nodes and intrapulmonary metastases with an optimal cut-off of 6.03 ng/mL for this purpose. Tumor size, subtypes, patients’ age and patients’ gender didn’t have any impact on the diagnostic performance of serum CEA levels in assessing metastasis. These results suggested that serum CEA was a valuable marker in assessing tumor metastasis in lung cancer patients.

Tumor metastasis, commonly occurred at stage IV, is responsible for majority of lung carcinoma death. Within the different types of lung carcinomas, there is also a preferential metastatic location, such as liver metastasis in SCLC and adenocarcinoma (Shin et al., 2014; Tamura et al., 2015). Most postoperative recurrences of NSCLC are distant metastasis, especially for brain metastasis and bone metastasis. Regardless of the subtype of lung cancer in the patient, the location and quantity of metastasis have a great impact on the selection of clinical treatment options and early intervention. Previous studies have reported that high levels of serum CEA or cerebrospinal fluid CEA are closely related to brain metastasis of lung cancer because of the capacity of CEA to penetrate the blood–brain barrier, behaving in a similar manner to immunoglobulins due to their homologous molecular weights. Similarly, studies also indicated that the expression of serum CEA is associated with bone metastasis in NSCLC (Li et al., 2016; Noris-Garcia & Escobar-Perez, 2004; Reiber, 2001; Reiber, Jacobi & Felgenhauer, 1986). Our findings also paid attention to the significant increase of serum CEA in lung cancer patients with multiple organ metastases. In addition, patients with distant metastasis showed notably higher CEA expression compared with mediastinal/hilar lymph node metastasis and intrapulmonary metastasis patients. Treatment and long-term outcomes depend on the type, stage, and general condition of cancer. Common treatments include surgery, chemotherapy and radiotherapy. NSCLC is sometimes treated by surgery, while SCLC usually has a better response to chemotherapy and radiotherapy. Early identification of the presence and location of metastasis is helpful for clinical selection of appropriate treatment options, such as minimally invasive surgery, palliative surgery, radiotherapy and bone modifier therapy can create better prognosis thus benefiting patients (Reck et al., 2014). High costs of examination, lack of obvious clinical symptoms and other factors can make it difficult to perform comprehensive examinations (such as whole-body imaging) to find and confirm the presence of metastases. But efforts to timely and comprehensively identify tumor metastasis in lung cancer patients would help patients choose rational therapies and prolong the survival of patients. Therefore, our study suggested the serum CEA as a valuable marker for tumor metastases assessment in newly diagnosed lung cancer patients, which could have some implications in clinical application. If elevated serum CEA is found in newly admitted patients with highly suspected lung cancer, more detailed clinical diagnostic measures should be actively sought to improve the timeliness and comprehensiveness of the detection of metastasis.

Limitations

Several limitations in our study should be acknowledged. Firstly, it was a retrospective-design single center study, which may make the results less generalization. Secondly, the time period of patients retrospectively reviewed was only one year, and the sample size was relatively small.

Conclusions

Our study revealed the significant qualitative difference of serum CEA in lung cancer patients with or without tumor metastasis and evaluated the diagnostic performance of CEA testing to determine tumor metastasis. In metastasized cases, patients with multiple metastases had significantly higher CEA levels compared to those with single metastasis; distant metastasis patients showed significantly higher CEA compared with non-distant metastasis cases. Furthermore, tumor size, subtypes, patients’ age and gender have no impact on the performance of CEA testing. Taken together, serum CEA has a positive clinical application value in combined with existed techniques for diagnosis of cancer metastasis to timely and comprehensively detect metastasis.

Supplemental Information

Dataset S1 Raw data

Click here for additional data file.

Figure S1 The location and distribution of lung cancer metastasis in different histological types

Click here for additional data file.

Figure S2 Imaging evidence for tumor metastases

(A) 1: Brain metastasis. (B) 2: Rib metastasis, 3: Massive pleural effusion. (C) 4: Pyramidal bone metastasis. (D) 5: Bone metastasis.

Click here for additional data file.

Figure S3 Relation between log(CEA) level and metastasis without predefined model structure

Click here for additional data file.

Figure S4 Impacts of histology, tumor size, patients age and gender on the performance of CEA.

(A) Crude ROC curve versus ROC curve adjusted for histological diagnosis. (B) tumor size. (C) age and (D) gender (outcome = metastasis).

Click here for additional data file.

Figure S5 Scatterplot of tumor size and CEA levels (Multiple R2 = 0.001, Adjusted R2 =  − 0.005, p-value = 0.68)

Click here for additional data file.

Additional Information and Declarations

Competing Interests

Author Contributions

Human Ethics

Data Availability

The authors declare there are no competing interests.

Jiasi Wang conceived and designed the experiments, performed the experiments, approved the final draft.

Yanpeng Chu, Xiangdong Fang and Fanwei Zeng analyzed the data, prepared figures and/or tables, approved the final draft.

Jie Li performed the experiments, authored or reviewed drafts of the paper, approved the final draft.

Tingjie Wang, Liangli Sun and Pingfei Wang authored or reviewed drafts of the paper, approved the final draft.

Junfeng Wang designed the analysis, analyzed the data, contributed reagents/materials/analysis tools, prepared figures and/or tables, authored or reviewed drafts of the paper, approved the final draft.

Fanxin Zeng conceived and designed the experiments, performed the experiments, analyzed the data, prepared figures and/or tables, authored or reviewed drafts of the paper, approved the final draft.

The following information was supplied relating to ethical approvals (i.e., approving body and any reference numbers):

The medical ethics review board of Dazhou Central Hospital approved to carry out the study within its facilities (Ethical Application Ref: IRB00000002-17002).

The following information was supplied regarding data availability:

The raw measurements are available in Dataset S1. The raw data shows all basic data for statistical analysis.

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
