# Peer review of "The clinical value of carcinoembryonic antigen for tumor metastasis assessment in lung cancer"

_PeerJ, doi:10.7717/peerj.7433_

## Round 0.1 · original submission · Major Revisions

While your manuscript addresses an important scientific question related to the diagnosis of metastatic lung cancer, there are several issues which need to be addressed before considering for publication in our journal. Please address the reviewer comments on a point to point basis and make changes to your manuscript accordingly.

[]

Reviewer 1 ·

Basic reporting

This study addresses the findings that patients with metastatic lung cancer have increased levels of carcinoembryonic antigen or CEA in the serum. On the basis of this finding, they propose CEA as a diagnostic marker for metastatic lung cancer. The background literature is well presented where the authors address the prevalence of lung cancer, detection limits, types of lung cancers, sites of metastasis and diagnostic methods together with their caveats, opening the foray for establishing the knowledge/clinical gap – need for economically viable convenient and non-invasive/non-harmful techniques to identify lung cancer progression and metastasis. The authors also explain why CEA is a diagnostic marker of choice for NSCLC with a range of detection.

The language is largely clear and umambiguous. Some examples of grammatical errors include:
1. Line 129: “Serum CEA was significantly higher in the metastatic group”.
2. Choice of words: rather than saying “risk of having metastasis” the authors can consider saying “developing metastasis”. Ditto for line 178.
3. Line 181: “which closed to the…”. Do the authors mean “which was close to the…”?
4. Unnecessary use of “the”. Example, Line 183: “Besides, in patients with tumor metastasis…” and Line 184 “in patients with multiple metastases compared with patients with single metastasis”.
5. Line 189: what is a “vulnerable” indicator?

The article is mostly professionally presented with tables and figures. Few comments on the figures:
1. P-values for all the box plots are missing from legends and/or the plots themselves.
2. The boxplot lines need to be capped on top and bottom

Experimental design

The manuscript falls within the scope of the journal.
While the research question is well-defined in that the authors are endeavoring to establish the diagnostic robustness of CEA in lung cancers, there are some concerns as follows:

1. Lines 73 and 76: Both these studies show high levels of CEA are associated with worse survival of lung cancer or development of brain metastasis in patients with advanced lung cancer. What specific findings does this study contribute that sets them apart from these other studies? The authors claim that “in-depth study on the relationship between serum CEA and tumor metastasis in cancer is still insufficient”, however, it is unclear what this specific analysis is that distinguishes this study from previous analyses.
2. CEA as a cancer marker has been around for decades. What is the probability of false positives for CEA in serum? Can stress or injury or other non-cancerous situations also increase CEA in blood/serum?
3. Where did the metastasis occur? If the metastasis occurred in the brain, how was this detected? In other words, can there be patients in the non-metastatic cohort that had “undetected” metastasis? For example, if CEA is really elevated in patients with metastasis, then the high data scatter above the range of 5 (log CEA) in Figure 2 non-metastasis group might actually have undetected metastasis.
4. Line 131: “risk of having metastasis” – was this done with the patients who had already developed metastasis? If so, how is risk predictive in already metastatic patients?
5. I am surprised that tumor size has no predictive correlation with CEA levels, given that metastasis occurs when tumors proliferate and outgrow their current niche and need to find secondary niches in which to establish themselves. Since CEA is a predictive marker for metastasis, it is interesting that tumor size has no bearing on this.

My major concerns are to do with the (A) premise of the study and (B) robustness of the data

(A) The point of a diagnostic marker is that cancer/metastasis is detected early and thereby prevented. However, according to the authors, at first diagnosis, up to 70% of the patients are already in advanced stages of cancer”. So, how do the authors plan to reduce this percentage if the patients do not present with any symptoms until it’s too late? Only when patients present symptoms, will they be analyzed for CEA levels, was that to become a tractable diagnostic tool. It would be useful for the authors to specify what diagnostic benefit are the authors proposing here? Are the authors suggesting the use of CEA detection in addition to surgery and imaging techniques? Or that CEA could replace imaging? The study seems like a redundant attempt, since the authors do not show whether CEA detection is better than existing methods such as CT-scanning, PET imaging etc.

(B)The authors state in lines 210-212 that CT screening has a lot of false positives. However, the data that they included in their analyses includes the determination of whether or not metastasis was detected based on CT scanning. How did the authors ensure that the false positives were eliminated?

Validity of the findings

Data seems statistically sound however the authors need to indicate the p-values in the plots. Conclusions are well stated, linked to the original research questions and encompass the results.

Reviewer 2 ·

Basic reporting

The manuscript consists of lots of grammatical errors which hinders the comprehension. The authors should address this.

The authors use professional article structure including the figures and tables.

Experimental design

In line 105 – 106: The authors state that tumor metastases were mainly diagnosed by imaging modalities (CT, and/or PET) or surgery within one month after hospitalization. How did the authors manage to get good resolution to identify single mets, especially in the organs such as brain. the authors should present data on CT/PET imaging.

In line 124: The authors demonstrate the patients’ demographic characteristics in Table 1. What is the relevance of these analysis in this study?

How did the authors divide the 213 tumors into 128 patients with metastasis and 85 patients without metastasis? The authors should demonstrate data in support of this.

In line 150: How did the authors divide the metastasis into distant metastasis (defined as Distant metastasis group) and 21 patients with regional tumor metastasis only?

Validity of the findings

In line 210: The authors state that CT screening is associated with a high percentage of false positive tests, which may lead to unnecessary treatments. However they use the same method to identify metastasis.

---

## Round 0.2 · accepted · Accept

Your manuscript has been successfully revised according to the reviewer's comments.

Reviewer 1 ·

Basic reporting

The authors have made satisfactory changes to the manuscript and I have no further comments and/or recommendations

Experimental design

The authors have made satisfactory changes to the manuscript and I have no further comments and/or recommendations

Validity of the findings

The authors have made satisfactory changes to the manuscript and I have no further comments and/or recommendations

Additional comments

The authors have made satisfactory changes to the manuscript and I have no further comments and/or recommendations